# Rethinking Deep Spiking Neural Networks: A Multi-Layer Perceptron Approach

## Abstract

By adopting deep convolution architectures, spiking neural networks (SNNs) have recently achieved competitive performances with their artificial counterparts in image classification, meanwhile with much lower computation cost due to event-driven and sparse activation. However, the multiplication-free inference (MFI) principle makes SNNs incompatible with attention or transformer mechanisms which have shown significant performance gains on high resolution vision tasks. Inspired from recent works on multi-layer perceptrons (MLPs), we explore an efficient spiking MLP design using batch normalization instead of layer normalization in both the token and the channel block to be compatible with MFI. We further strengthen the network's local feature learning ability with a spiking patch encoding layer, which significantly improves the network performance. Based on these building blocks, we explore an optimal skip connection configuration and develop an efficient multi-stage spiking MLP network combining global receptive field and local feature extraction, achieving full spike-based computation. Without pre-training or other advanced SNN training techniques, the spiking MLP network achieves 66.39% top-1 accuracy on the ImageNet-1K dataset, surpassing the state-of-the-art directly trained spiking ResNet-34 by 2.67% under similar model capacity meanwhile with shorter simulation steps and much less computation cost. Another larger variant of the network achieves 68.84% top-1 accuracy, rivaling the spiking VGG-16 network with 4 times smaller model capacity. Our work demonstrates the effectiveness of an alternative deep SNN architecture combining both global and local learning abilities. More interestingly, finally we show a close resemblance of the trained receptive field of our network to cells in the cortex. `Code_will_be_publicly_available.`

## 1 Introduction

Spiking neural networks (SNNs) (Maass, 1997) have been proposed as models for cortical simulation (Izhikevich, 2004; Brette & Gerstner, 2005; Deco et al., 2008; Gerstner et al., 2014; Korcsak-Gorzo et al., 2022) and candidates for solving problems in machine learning (Tavanaei et al., 2019; Roy et al., 2019). Nevertheless, SNNs following exact biological topology and constraints such as Dale's law have not been demonstrated to be equally effective as artificial neural networks (ANNs) in practice, especially when scaling up. By adopting structures and adapting learning algorithms from their artificial counterparts, SNNs have improved their performances and recently achieved higher accuracy in benchmark image classification problems (Shrestha & Orchard, 2018; Wu et al., 2019; Sengupta et al., 2019; Li et al., 2021; Fang et al., 2021; Deng et al., 2022). Deep convolutional neural networks (CNNs) are the current de-facto architectures adopted by SNNs in various vision tasks (Kim et al., 2020; 2021; Rançon et al., 2021; Zhu et al., 2022). Recently, ANNs with visual attention and transformer mechanisms (Dosovitskiy et al., 2020; Liu et al., 2021) have surpassed pure CNNs by learning global dependency of the image. However, these mechanisms usually involve matrix multiplication and softmax function which make them contradict to the multiplication-free inference (MFI) principle of SNNs (Roy et al., 2019; Rathi & Roy, 2021).

In this work, we explore a spike-based implementation of an alternative structure more compatible with this principle, i.e multi-layer perceptrons (MLPs), which have recently been demonstrated to be equally efficient as transformers (Tolstikhin et al., 2021). The original MLP-Mixer architecture for ANNs still involves real-valued matrix multiplication which violates MFI. To this end, we design a

spiking MLP-Mixer architecture using the MFI-friendly batch normalization (BN) with lightweight axial sampling in the token block. With the spiking MLP-Mixer as a basic building block, we propose a multi-stage spiking-MLP network achieving full spike-based computation. To enhance the local feature extraction of the MLP network, we propose a spiking patch encoding module based on directed acyclic graph structure to replace the original patch partition for downsampling. In addition, we identify the crucial role of skip connection configuration for an optimal spiking MLP-Mixer design. To our best knowledge, this is the first work to explore full spike-based token-sampling MLP architectures in the field of SNNs. To be specific, our contributions can be summarized as follows:

- We develop an efficient spiking MLP-Mixer with MFI-friendly BN and lightweight axial sampling in the token block. In addition, we identify the crucial role of skip connection configuration for an optimal spiking MLP-Mixer design.

- We propose a spiking patch encoding module to enhance local feature extraction and for downsampling, based on which we construct a multi-stage spiking-MLP network achieving full spike-based computation.

- Our network achieves 66.39% top-1 accuracy on the classification of ImageNet-1K with 2.67% improvement compared to the current state-of-the-art deep spiking ResNet-34 network, meanwhile with similar model capacity, 2/3 of its simulation steps and much lower computation cost. With an equal simulation steps the same network improves to 69.09% accuracy achieving 5.37% improvement, slightly surpassing spiking VGG-16 network with 5.5 times smaller model capacity, demonstrating the effectiveness of an alternative architecture design for deep SNNs.

- Finally, our networks pre-trained on ImageNet create new SNN records when fine-tune on CIFAR10 and CIFAR100 datasets, achieving 96.08% and 80.57% accuracy, demonstrating the general usage of our architecture as pre-trained models.

## 2 RELATED WORK

### 2.1 SPIKING NEURAL NETWORKS IN DEEP LEARNING

Originated from computational neuroscience, SNNs have been widely used for modeling brain function and dynamics (Izhikevich, 2004; Brette & Gerstner, 2005; Deco et al., 2008; Gerstner et al., 2014; Korcsak-Gorzo et al., 2022). The success achieved and challenges faced by deep ANNs in solving machine learning problems have led a trend to use SNNs as an alternative and explore functional benefits of their bio-inspired properties in solving similar problems. It has been demonstrated that by adapting learning algorithms (Bohte et al., 2000; Wu et al., 2018; Neftci et al., 2019; Bellec et al., 2020) and adopting efficient architectures from ANNs, such as Boltzmann machines (Ackley et al., 1985) in the early phase of deep learning and the current dominant CNNs (LeCun et al., 1989), SNNs can achieve competitive performances rivaling their artificial counterparts (Petrovici et al., 2016; Neftci et al., 2014; Leng et al., 2018; Shrestha & Orchard, 2018; Wu et al., 2019; Zhang & Li, 2020; Li et al., 2021; Fang et al., 2021; Deng et al., 2022). Several recent works have applied neural architecture search (NAS) to obtain task-specific cells or network structures for SNNs (Na et al., 2022; Kim et al., 2022). Despite that these networks have achieved further spike reduction and new state-of-the-art results in image classification, their architectures are still deep CNNs.

### 2.2 BIOLOGICALLY PLAUSIBLE ATTENTION

The recent success of attention and transformer mechanisms in speech (Vaswani et al., 2017) and vision tasks (Dosovitskiy et al., 2020) has motivated explorations of similar mechanisms in more biologically plausible forms. Works in Widrich et al. (2020); Ramsauer et al. (2020) showed that the attention mechanism of transformer is equal to the update rule of modern Hopfield networks with continuous states and demonstrated its high storage capacity in large scale multiple instance learning. However, same as the original transformer, it involves matrix multiplication and softmax function which are not compatible with spike-based computation. The pioneering work of MLP-Mixer (Tolstikhin et al., 2021) proposed an alternative architecture based exclusively on MLPs without convolution or self-attention layers. The approach contains two MLP blocks with a token mixing block applied on sliced image patches similar to vision transformer (ViT) (Dosovitskiy et al., 2020)

and a channel mixing block applied across patches. Without the inductive biases of the local connectivity and the self-attention, the token-mixing MLP is more flexible and has a stronger fitting capability (Zhao et al., 2021; Liu et al., 2022). However, it is also more prone to over-fitting and relies on pre-trained models on large-scale datasets to achieve competitive performances on image classification benchmarks comparable to ViT and CNNs. Tang et al. (2022) proposed sparseMLP to solve the over parametric and over-fitting problem of MLP-Mixer by adopting a multi-stage pyramid network structure and applying axial sampling instead of full sampling in the token mixing MLP. Similar concepts were also shared in other MLP variants (Hou et al., 2022; Tatsunami & Taki, 2021; Wang et al., 2022). The structural simplicity and effectiveness of MLPs suggest a promising network paradigm. Li et al. (2022) implemented MLPs with non-spiking full-precision neurons. Nevertheless, the network is not a typical SNN since it avoids spike-based communication and fails to fulfill the MFI principle.

## 3 METHOD

### 3.1 TRAINING OF SNN

We adopt the leaky integrate-and-fire (LIF) neuron model with hard threshold and decay input described by

$$\boldsymbol{u}^{t,\mathrm{pre}} = \boldsymbol{u}^{t-1} + \frac{\boldsymbol{z}^t - \boldsymbol{u}^{t-1}}{\tau} \tag{1}$$

$$\boldsymbol{y}^t = g(\boldsymbol{u}^{t,\mathrm{pre}}) \tag{2}$$

$$\boldsymbol{u}^t = (1 - \boldsymbol{y}^t)\boldsymbol{u}^{t,\mathrm{pre}} \tag{3}$$

where $t$ denotes the time step, $\tau$ is the membrane time constant, $\boldsymbol{u}$ is the membrane potential denoted by bold italic letter representing a vector, $\boldsymbol{y}$ denotes the spike output, $g$ is a threshold function, $\boldsymbol{z}^t = \boldsymbol{W}\boldsymbol{y}^{t,\mathrm{pre}}$ is the synaptic input with $\boldsymbol{W}$ denoting the weight matrix, and $\boldsymbol{y}^{t,\mathrm{pre}}$ denotes the afferent spikes from pre-synaptic neurons. A neuron will fire a spike and transmit it to the post-synaptic neuron when its membrane potential exceeds a threshold $V_{th}$, followed by a hard reset of the membrane potential; otherwise the neuron transmits no signals, given by

$$y_i^t = \begin{cases} 1 & \text{if } u_i^{t,\mathrm{pre}} \geq V_{th} \\ 0 & \text{otherwise} \end{cases} \tag{4}$$

In this work, we set $\tau = 2$ and $V_{th} = 1$. Given loss $L$ and using chain rule, the weight update of an SNN can be expressed as:

$$\frac{\partial L}{\partial \boldsymbol{W}} = \sum_t \frac{\partial L}{\partial \boldsymbol{y}^t} \frac{\partial \boldsymbol{y}^t}{\partial \boldsymbol{u}^{t,\mathrm{pre}}} \frac{\partial \boldsymbol{u}^{t,\mathrm{pre}}}{\partial \boldsymbol{z}^t} \frac{\partial \boldsymbol{z}^t}{\partial \boldsymbol{W}} \tag{5}$$

where $\frac{\partial \boldsymbol{y}^t}{\partial \boldsymbol{u}^{t,\mathrm{pre}}}$ is the gradient of the firing function, which is zero everywhere except at the threshold. The surrogate gradient method uses continuous functions to approximate the gradients, such as rectangular (Zheng et al., 2021), triangular (Bellec et al., 2018), exponential curves (Shrestha & Orchard, 2018; Zenke & Ganguli, 2018), etc. We adopt sigmoid function in our experiments.

### 3.2 NETWORK ARCHITECTURE

The MLP-Mixer (Tolstikhin et al., 2021) adopted a single-stage or "isotropic" network design where input and output resolutions were kept the same across different layers. This structure leads to excessive number of parameters which can cause an over-fitting of the network when training on medium-scale dataset such as ImageNet-1K. To alleviate this issue, we adopt a multi-stage pyramid network architecture as Liu et al. (2021); Chu et al. (2021); Tang et al. (2022). Figure 1 illustrates the architecture of our network.

**Spiking Patch Encoding** Given an input RGB image with size 3×H×W, the original MLP-Mixer divides it into non-overlapping patches of size $p \times p$ and linear projects the new channel dimension of $3p^2$ to a hidden dimension of $C_1$. This process is equal to a 2D convolution operation with kernel and stride size of $p$ and output dimension of $C_1$. Compared to convolution operation, the MLP has a

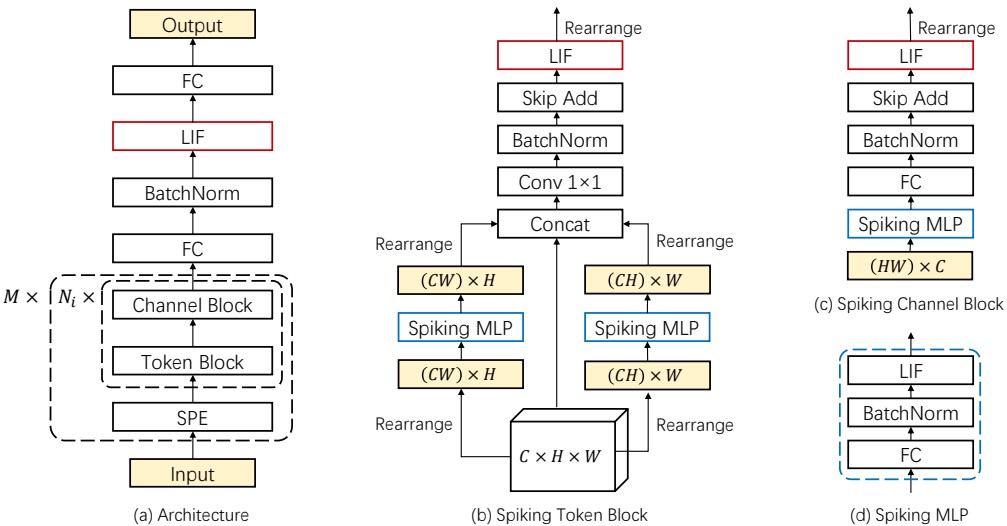

(a) Architecture  (b) Spiking Token Block  (c) Spiking Channel Block  (d) Spiking MLP

Figure 1: Overall network architecture (a) and the spiking MLP-Mixer (b-d). The multi-stage network is downsampled with a spiking patch encoding (SPE) module at each stage. Within each stage, the SPE is followed by a sequential of spiking MLP-Mixers with identical architecture each containing a spiking token block with axial sampling (b) and a spiking channel block with full sampling (c). For simplicity, we denote the repeated FC-BatchNorm-LIF block as a spiking MLP layer (d).

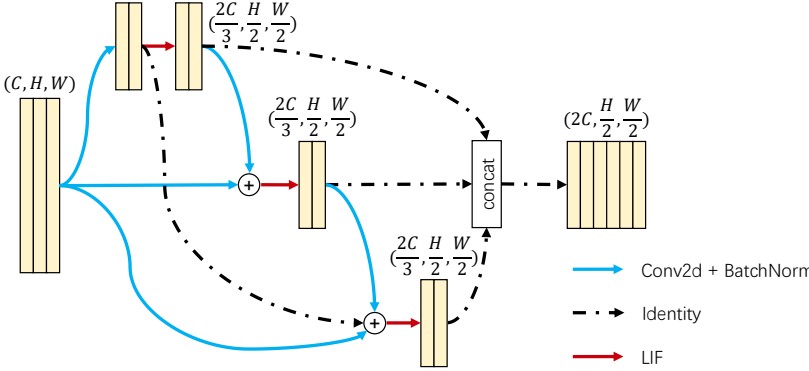

Figure 2: Spiking patch encoding with a directed acyclic graph structure. The structure follows MFI principle with additions performed on BN states and multiplications performed between convolution weights and binary spikes.

global receptive field but in turn lacks the inductive bias to learn from local features. To address this issue, we construct a spiking patch encoding (SPE) module to replace the original patch partition approach. The SPE is implemented with a spike-based directed acyclic graph inspired from a recent work (Anonymous), as shown in Figure 2. We call this structure a spiking cell since it is originated from DARTS (Liu et al., 2018). A cell consists of 3 nodes, where each node receives the same input from the previous stage with convolution operation followed by BN. In addition, node pairs (1,2) and (2,3) are interconnected with convolution followed by BN, and node pair (1,3) is connected with identity connection. Within each node, multiple operations are added after BN, and the following spiking activation is used as output of the node. The output of the cell is the concatenated spiking states of all nodes.

The output of the spiking cell is then fed into a sequential of spiking MLP-Mixers which finally output a spiking feature map of the same shape. In the following stages, this process is repeated

with the input feature consecutively downsampled in spatial dimension and expanded in channel dimension, both with a ratio of 2. The output of the final stage is fed into a spiking classification head with a spiking MLP block consisting of a fully connected (FC) layer followed by BN and a spiking activation, with the FC layer performing full sampling on the flattened spatial dimension of the feature map. Finally, a linear classifier projects the output of spiking MLP block to a label layer. The loss function is a cross entropy function with the network output averaged over simulation time steps.

### 3.3 SPIKING MLP-MIXER

A direct adoption of existing design of MLP blocks in ANNs by replacing the real-valued activation function with spiking activation function (SAF) leads to a contradiction to MFI principle of spike-based computation. In both of the token and channel block of the original MLP-Mixer, the FC operation is performed on the layer normalization (LN) state of the feature map, leading to real-valued matrix multiplication. To address this issue, we relocate the normalization after the FC operation and replace LN with BN, such that the parameters of the latter can be merged into linear projection weights during inference to be compatible with the MFI principle (Ioffe & Szegedy, 2015; Rueckauer et al., 2017).

A spiking MLP-Mixer consists of a token block and a channel block, each includes multiple FC layers with spiking activation, as illustrated in Figure 1b,c.

**Spiking Token Block** Given input image patches $\boldsymbol{I}$ of shape $C \times \frac{H}{p} \times \frac{W}{p}$, the token block of the original MLP-Mixer fattens its spatial dimension to form a 2D tensor and performs full sampling on the token dimension with weights shared across the channel dimension, thus leading to a global receptive field which can be heavily parameterized given small patch sizes. Inspired from previous works (Tatsunami & Taki, 2021; Hou et al., 2022; Tang et al., 2022; Wang et al., 2022), we design a two branch structure for the spiking token block to separately encode the feature representation along the horizontal and vertical spatial dimension which learns long-range dependencies along one direction meanwhile preserving positional information along the other direction. As shown in Figure 1c, each branch consists of an FC layer followed by a BN and an SAF. Afterwards, the two binary feature maps are concatenated along the channel dimension together with an identity connection from the input feature map, and projected back to $C$ through an FC layer with BN and an SAF. Additionally, here we add a skip connection from the BN states of the SPE.

**Spiking Channel Block** The channel block consists of two spiking FC layers and receives the transposed output from the token block. The first layer expands the channel dimension by a certain ratio $\alpha$ and the second layer recovers to the original dimension. A skip connection is added between the BN state of the second second layer and the summed BN state of the token block.

To alleviate the gradient vanish problem of deep SNN, we add skip connections between the last BN states of the token and the channel block, together with the BN states of the SPE. We show in ablation studies that this design is crucial for an efficient network. Within a stage, denoting matrix multiplications by two bold capital letters, the first spiking MLP-Mixer can be formulated as:

$$\boldsymbol{X} = \text{Concat}[\boldsymbol{X}_{n1}, \boldsymbol{X}_{n2}, \boldsymbol{X}_{n3}] \tag{6}$$

$$\boldsymbol{I}_1 = f(\boldsymbol{X}) \tag{7}$$

$$\boldsymbol{U}_h = \text{BN}(\boldsymbol{W}_h \boldsymbol{I}_1^h) \tag{8}$$

$$\boldsymbol{U}_w = \text{BN}(\boldsymbol{W}_w \boldsymbol{I}_1^w) \tag{9}$$

$$\boldsymbol{Y} = \text{BN}(\boldsymbol{W}_f \text{Concat}[f(\tilde{\boldsymbol{U}}_h), f(\tilde{\boldsymbol{U}}_w), \boldsymbol{I}_1]) + \boldsymbol{X} \tag{10}$$

$$\boldsymbol{I}_2 = f(\boldsymbol{Y}) \tag{11}$$

$$\boldsymbol{V} = \boldsymbol{Y}^c + \text{BN}(\boldsymbol{W}_{c2} f(\text{BN}(\boldsymbol{W}_{c1} \boldsymbol{I}_2^c))) \tag{12}$$

$$\boldsymbol{I}_3 = f(\tilde{\boldsymbol{V}}) \tag{13}$$

where BN denotes batch normalization, $\boldsymbol{X}_{ni}$ is the state before the spiking activation of the $i$th node, i.e summed up values of multiple combined operations, $f$ denotes an SAF, $\boldsymbol{W}_h$ and $\boldsymbol{W}_w$ are the token FC weights operating on the height and width dimension with shape $H \times H$ and $W \times W$ respectively, $\boldsymbol{W}_f$ is the branch fusion weights with shape $C \times 3C$, superscripts $h$, $w$ and $c$ denote the matrix reshape operation preserving the height, width and channel dimensions with $\boldsymbol{I}_1^h$, $\boldsymbol{I}_1^w$

and $\boldsymbol{I}_2^c$ of shape $H \times \frac{CW}{p^2}$, $W \times \frac{CH}{p^2}$ and $C \times \frac{HW}{p^2}$ respectively, ~ denotes the reshape operation flattening the spatial dimension with $\tilde{\boldsymbol{U}}_h$ and $\tilde{\boldsymbol{U}}_w$ of shape $C \times \frac{HW}{p^2}$, $\boldsymbol{W}_{c1}$ and $\boldsymbol{W}_{c2}$ are the channel FC weights with shape $\alpha C \times C$ and $C \times \alpha C$ respectively, where $\alpha$ is the channel expansion ratio. Note that all matrix multiplications are between a real-valued matrix and a binary matrix, such that the spiking MLP-Mixer is essentially multiplication-free. For the following spiking MLP-Mixers, the input tensor becomes the output from the channel block of the previous Mixer.

## 4 EXPERIMENTS

We evaluate our model on benchmark image classification datasets of ImageNet-1K (Krizhevsky et al., 2017) and CIFAR10/100 (Krizhevsky et al.). We use SpikingJelly (Fang et al., 2020) with its cupy backend to speed up the simulation of LIF neuron. In the SPE module, for convolution operations we use a kernel size of $3 \times 3$ with stride 2 and 1 for input to node and node to node connections, respectively.

### 4.1 IMAGENET-1K

The ImageNet-1K dataset consists of a training set, a validation set and a test set with 1.28M, 50K and 100K 224×224 images respectively. Most of our training settings are inherited from Tang et al. (2022) including data augmentation. We use cosine decay learning rate strategy. The initial learning rate is 0.1 and gradually drops to 0 in 100 epochs. The optimizer is SGD with a momentum of 0.9. For the first stage of the spiking MLP network we use the original patch partition approach with patch size 4 and the rest stages with SPE module for downsampling. We build three variants of spiking MLP network, namely spiking MLP-SPE-T/S/B. The architecture of these two models are: spiking MLP-SPE-T: $C_1 = 78$, number of layers in each stage = {2; 8; 14; 2}, spiking MLP-SPE-S: $C_1 = 96$, number of layers in each stage = {2; 8; 14; 2}, spiking MLP-SPE-B: $C_1 = 108$, number of layers in each stage = {2; 10; 24; 2}. The expansion ratio of the channel FC layer is set to $\alpha = 3$. To study the influence of the SPE module, we also train an MLP-S model with the original patch partition approach while other parts of the network remaining the same. The simulation step of SNN is set to $T = 4$. To better compare with other SNNs with larger time steps, we use time inheritance training (TIT) (Deng et al., 2022) and obtain the $T = 6$ result for MLP-SPE-T, where we initialize the network with the pre-trained MLP-SPE-T ($T = 4$) model and fine-tune for 50 epochs with $T = 6$ using a cosine learning rate decaying from 0.1 to 0. We compare our model with other state-of-the-art SNNs including directly training and ANN-to-SNN conversion methods. The result is in table 1 where we report the top-1 accuracy along with the number of model parameters and simulation steps. We can see that ANN-SNN conversion methods can achieve

Table 1: Comparison on ImageNet-1K. T denotes simulation length.

| Method | Architecture | Model Size | T | Accuracy[%] |
|---|---|---|---|---|
| ANN-SNN (Hu et al., 2018) | ResNet-34 | 22M | 768 | 71.6 |
| ANN-SNN (Sengupta et al., 2019) | VGG-16 | 138M | 2500 | 69.96 |
| Hybrid training (Rathi et al., 2020) | ResNet-34 | 22M | 250 | 61.48 |
| Hybrid training (Lee et al., 2020) | VGG-16 | 138M | 250 | 65.19 |
| STBP-tdBN (Zheng et al., 2021) | ResNet-34 | 22M | 6 | 63.72 |
| TET (Deng et al., 2022) | ResNet-34 | 22M | 6 | 64.79 |
| STBP-tdBN (Zheng et al., 2021) | ResNet-34-large | 86M | 6 | 67.05 |
| Diet-SNN (Rathi & Roy, 2021) | VGG-16 | 138M | 5 | 69.00 |
| **Spiking MLP** (our model) | MLP-SPE-T | **25M** | **4** | **66.39** |
| **Spiking MLP** (our model) | MLP-SPE-T | **25M** | **6** | **69.09** |
| **Spiking MLP** (our model) | MLP-S | **34M** | **4** | **63.25** |
| **Spiking MLP** (our model) | MLP-SPE-S | **38M** | **4** | **68.84** |
| **Spiking MLP** (our model) | MLP-SPE-B | **66M** | **6** | **71.64** |

high accuracy however with extremely long simulation steps. Hybrid training methods reduce the simulation steps but are still significantly longer than directly training methods. Among directly trained methods, under similar network capacity, MLP-SPE-T significantly surpasses STBP-tdBN

ResNet-34 by 2.67% with shorter simulation time steps. After TIT the model achieves an accuracy of 69.09% with $T = 6$, surpassing ResNet-34-large and VGG-16 model by 2.04% and 0.09% with more than 3 and 5.5 times smaller model capacity, demonstrating the superiority of our architecture. The improvement of MLP-SPE-S from MLP-S shows that the SPE module can significantly improve network performance, demonstrating the effectiveness of combining global receptive field and local feature extraction. Note that it is more fair to compare our methods with the STBP-tdBN method since we use the same conventional temporal averaged loss function. The TET method used a temporal moment-wise loss function that can outperform temporal averaged one as demonstrated in Deng et al. (2022).

## 4.2 CIFAR

Both the CIFAR10/100 datasets contain 50K training images and 10K testing images with a size of $32 \times 32$ pixels. We use random resized crop and random horizontal flip for data augmentation as other works. We adopt the pre-training and fine-tuning paradigm as other transformer and MLP works (Dosovitskiy et al., 2020; Tolstikhin et al., 2021). We use the pre-trained spiking MLP-SPE-T network on the ImageNet-1K dataset as the pre-trained model. We reset the final classification output layer of the network and fine-tune it on the resized $224 \times 224$ CIFAR10/100 dataset. For fine-tuning, we use a cosine decay learning rate with an initial value of 0.1 and gradually drops to 0 in 100 epochs. The optimizer is SGD with a momentum of 0.9. We compare the result with other state-of-the-art SNNs as shown in table 2. With a larger network size, our model sets new records on both datasets, significantly surpassing existing SNNs. It might be a bit unfair to compare with directly trained SNNs, nevertheless the results demonstrate the usefulness of the spiking MLP network as a pre-trained model for other datasets.

Table 2: Comparison on CIFAR. T denotes simulation length.

| Dataset | Method | Architecture | T | Accuracy[%] |
|---|---|---|---|---|
| CIFAR10 | Diet-SNN (Rathi & Roy, 2021) | ResNet-20 | 10 | 92.54 |
|  | STBP-tdBN (Zheng et al., 2021) | ResNet-19 | 6 | 93.16 |
|  | STBP-tdBN (Zheng et al., 2021) | ResNet-19 | 4 | 92.92 |
|  | TET (Deng et al., 2022) | ResNet-19 | 4 | $94.44 \pm 0.08$ |
|  | **Spiking MLP** (our model) | MLP-SPE-T | **4** | **96.08** |
| CIFAR100 | Diet-SNN (Rathi & Roy, 2021) | ResNet-20 | 5 | 64.07 |
|  | STBP-tdBN (Zheng et al., 2021) | ResNet-19 | 6 | $71.12 \pm 0.57$ |
|  | STBP-tdBN (Zheng et al., 2021) | ResNet-19 | 4 | $70.86 \pm 0.22$ |
|  | TET (Deng et al., 2022) | ResNet-19 | 6 | $74.72 \pm 0.28$ |
|  | TET (Deng et al., 2022) | ResNet-19 | 4 | $74.47 \pm 0.15$ |
|  | **Spiking MLP** (our model) | MLP-SPE-T | **4** | **80.57** |

## 4.3 ABLATION STUDY

In this section we further study the influence of different configurations of skip connections on the performance of the network. As shown in Figure 3, we distinguish different skip connections, i.e patch block to token block (PT), patch block to channel block (PC), token block to channel block (TC) and channel block to the next token block (CT), with corresponding indices and colors. The skip connection from the patch block to the first token block is set as default. To shorten the simulation time, we set the initial channel number to 60 and directly train networks of different skip connection configurations from scratch on the CIFAR10 dataset for 100 epochs. The results are collected in Table 3. Networks without skip connections from the initial patch block to sequential token and channel blocks perform much worse than the others. This could be due to long range skip connections alleviate the gradient vanish problem of deep SNNs during training. Our spiking MLP-mixer adopts the optimal skip connection configuration of (PT, PC, TC).

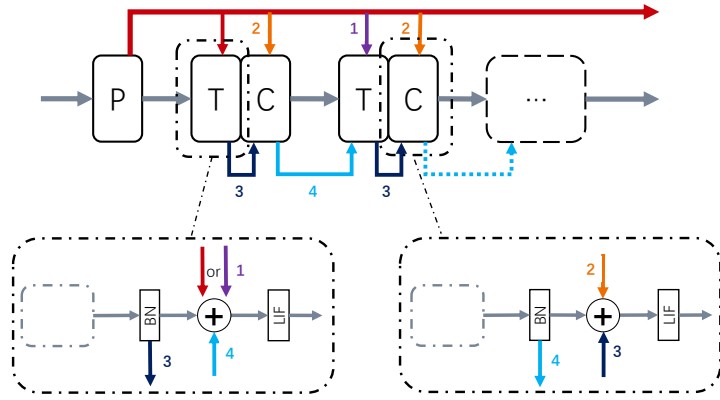

Figure 3: Potential skip connections for the spiking MLP-Mixer.

Table 3: Network performance under different skip connection configurations on CIFAR10.

| PT(1) | PC(2) | TC(3) | CT(4) | Accuracy[%] |
|:-----:|:-----:|:-----:|:-----:|:-----------:|
| ✓ | ✓ | | | 80.53 |
| ✓ | ✓ | ✓ | | 81.35 |
| ✓ | ✓ | ✓ | ✓ | 79.46 |
| | | ✓ | ✓ | 10.53 |
| | | | ✓ | 10.00 |

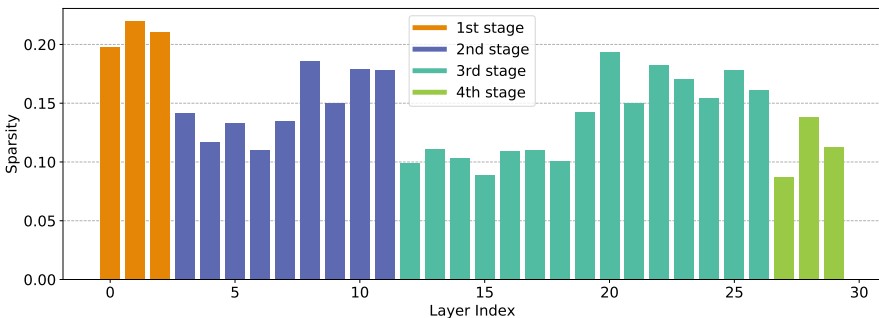

Figure 4: Mean network sparsity of spiking MLP-SPE-T on the ImagNet-1K test set. We distinguish each stage with different colors.

## 4.4 NETWORK SPARSITY AND COMPUTATION COST

SNNs are considered to be much more energy efficient than dense-computing ANNs, due to their inherent event-driven and sparse computing property. In addition to that, the MFI principle enables SNNs to be fully based on additions, which further decreases their potential energy consumption comparing with ANNs relying on real-valued matrix multiplications. In this section, we measure the sparsity as well as the computation cost of the spiking MLP-SPE-T model and compare with spiking ResNet-34. We plot the mean sparsity of each spiking MLP-Mixer on the ImageNet-1K test set in Figure 4. Our model shows varying sparse activity across different layers with a mean network sparsity of 0.14. Following (Li et al., 2021; Rathi & Roy, 2021), we count the number of addition operations of the SNN by $s * T * A$, where $s$ is the mean sparsity (total number of spikes divides total feature map size), $T$ is the simulation time step and $A$ is the addition number. The result is Table 4. The small amount of multiplication is from the input layer which receives floating-value images. We estimate the energy consumption of the network following the study of Horowitz (2014) on

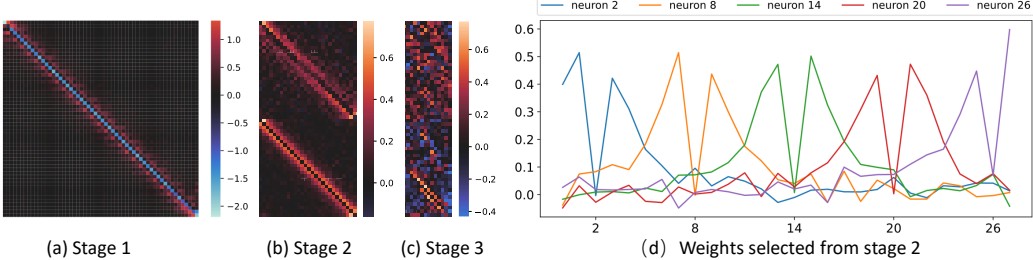

(a) Stage 1      (b) Stage 2      (c) Stage 3      (d) Weights selected from stage 2

Figure 5: Axial sampling weights from token blocks across different stages after training on ImageNet-1K. (a-c) Randomly selected token weights ploted in $H \times H$ or $W \times W$ 2D matrix. 1, 2 and 4 weight matrices from stage 1, 2 and 3 are displayed. The diagonal distribution of values in early stages indicates local sampling. (d) Weights of several neurons from the same token block in stage 2. For simplicity we only plot several neurons here, this 'M' shaped weight kernel is found universal in early stages of the network.

45nm CMOS technology adopted by previous works (Li et al., 2021; Rathi & Roy, 2021), with one addition operation in SNN costing $0.9$pJ and one multiply-accumulate (MAC) operation in ANNs consuming $4.6$pJ. With similar model capacity, our network has a computation cost almost twice smaller than spiking ResNet-34.

Table 4: The computation and estimated energy cost.

| Model | #Param. | Accuracy[%] | #Add. | #Mult. | Energy |
|---|---|---|---|---|---|
| Spiking ResNet-34 | 22M | 63.72 | 1.85G | 118M | 2.21 mJ |
| Spiking MLP-SPE-T | 25M | 66.39 | 1.18G | 12M | **1.12** mJ |

## 4.5 VISUALIZATION

Finally, we visualize weights from token blocks across different stages of spiking MLP-SPE-T after training on ImageNet-1K and plot them in Figure 5. Interestingly, it can be observed that weights in the early stage concentrate on local areas while those of latter stages gradually sample more global areas. Without inductive bias, the SNN naturally learns a hierarchically arranged receptive fields. In addition, receptive fields of spiking neurons in early stages resemble those 'off-center on-surround' receptive fields of cortex cells (Hubel & Wiesel, 1962), with a central inhibitory area surrounded by an excitatory area. This 'M' shaped weight kernel is found universal in early stages of the network. It would be interesting to see if similar phenomena can be observed in ANNs.

## 5 CONCLUSION

In this work, we constructed MLP architectures with spiking neurons following the MFI principle. The spiking MLP-Mixer adopted batch normalization instead of layer normalization in both the token and the channel block to be compatible with MFI. We further strengthened the network's local feature learning ability with a spiking patch encoding layer, which significantly improved the network performance. Based on these building blocks, we explored an optimal skip connection configuration and developed an efficient multi-stage spiking MLP network combining global receptive field and local feature extraction. Under similar model capacity, our networks significantly outperform state-of-the-art mainstream deep spiking convolutional networks on ImageNet-1K dataset in terms of a balance between accuracy, model capacity and computation cost, demonstrating the effectiveness of such an alternative architecture for deep SNNs. Our work suggests the importance of integrating global and local learning for optimal architecture design of SNN. Finally, the trained receptive fields of our network closely resemble those of cells in the cortex. A future comparison with ANNs would be interesting.

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

## A  APPENDIX

### A.1  DATA AUGMENTATION

For experiments on the ImageNet dataset, we inherited the default data augmentation from Tang
et al. (2022), which is different from the one He et al. (2016) used in SNNs we have compared
with (Zheng et al., 2021; Li et al., 2021; Deng et al., 2022). We denote the former as strong data
augmentation while the latter as simple data augmentation and perform the ablation study. The
results are summarized in Table 5 and the training curves of top-1 accuracy are plotted in Figure
6. Our model performs slightly better with simple data augmentation, probably due to a suboptimal
learning rate when we use strong data augmentation.

Table 5: Results comparison on ImageNet-1K using simple and strong data augmentation.

| Architecture | Data Augmentation | T | Accuracy[%] |
|---|---|---|---|
| MLP-SPE-T | Strong | 4 | **66.39** |
| MLP-SPE-T | Simple | 4 | **66.84** |

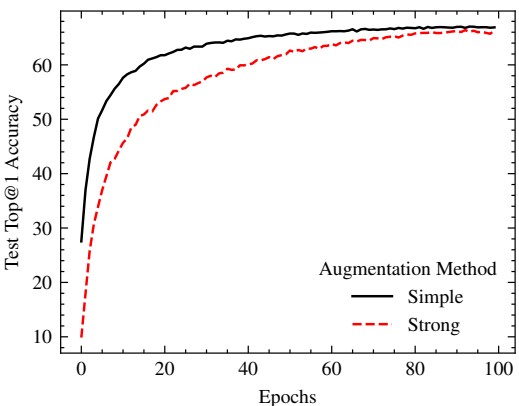

Figure 6: Training curves of top-1 accuracy of spiking MLP-SPE-T using simple and strong data augmentation.

