# OpenReview forum: "Rethinking Deep Spiking Neural Networks: A Multi-Layer Perceptron Approach"
_ICLR.cc/2023/Conference — Submitted to ICLR 2023_

### Official Review · Reviewer_8C6v · 2022-10-24

**Confidence:** 4
**Correctness:** 4
**Technical Novelty And Significance:** 3
**Empirical Novelty And Significance:** 3
**Recommendation:** 6

**Clarity, Quality, Novelty And Reproducibility:**

The authors say nothing about the FLOPs nor the throughput.

The authors are correct in saying that batch norm is not a pb for SNNs (can be absorbed in adjacent conv layers), but layer norm is (require multiplications at inference time, and in addition, it's not local)



**Strength And Weaknesses:**

STRENGTHS:

Accuracy on CIFAR10 and 100 is excellent.

WEAKNESSES:

The method is not competitive on ImageNet, suggesting that it does not scale well. Fang et al is cited, but does not appear in Table 1! Their SEW ResNet50 reaches 67.8% with 4 timesteps, vs 66.4% here.





**Summary Of The Paper:**

The authors train spiking neural networks (SNNs) for image classification using the surrogate gradient method. They propose a MLP architecture, where (strided) convolutions are only found in the patch encoding stages. The rest of the network uses linear layers only (no attention either). They report results on CIFAR10, CIFAR100 and ImageNet.

**Summary Of The Review:**

The method is competitive on CIFAR but not on ImageNet.

---

### Official Review · Reviewer_saxA · 2022-10-25

**Confidence:** 4
**Correctness:** 3
**Technical Novelty And Significance:** 2
**Empirical Novelty And Significance:** 2
**Recommendation:** 3

**Clarity, Quality, Novelty And Reproducibility:**

Clarity: 6/10
Quality: 6/10
Novelty: 4/10
Reproducibility: N/A but authors promised to release code after review.

**Strength And Weaknesses:**

Strength:

+ The authors ensure multiplication-free inference.

+ This work may be the first attempt for combining LIF and MLP.

Weakness:

- The motivation for this work is not convincing for me. In the abstract, the authors claimed that the current MLP does not support MFI and that is why they want to improve spikingMLP. For me, I think the real advantage of applying SpikingMLP is whether it is extremely highly efficient compared to SpikingCNN in training or inference; whether SpikingMLP has higher transferability than SpikingCNN; or whether SpikingMLP has less gap between ANN and SNN accuracy than SpikingCNN. The current work looks like a simple combination.

- Section 3.2 said  *This structure leads to excessive number of parameters which can cause an over-fitting of the network when training on medium-scale dataset such as ImageNet-1K. To alleviate this issue, we adopt a multi-stage pyramid network architecture as Liu et al. (2021); Chu et al. (2021); Tang et al. (2022).* This questions me again for the motivation of this paper. If simply applying the global architecture like MLP-Mixer the SNN will become much worse, does that mean spiking neurons are not suitable for MLP architecture (no inductive bias one) at all? More concretely, I am curious what is the compatibility of LIF neurons for MLP structure, is it strongly relied on inductive bias or it can learn inductive bias by itself?

- I think the accuracy cannot be interpreted very well by readers. For example, in the original MLP Mixer results table, there are inference throughput and training days compared to CNN and transformer. If possible, can authors also provide these two hardware performances since they are completely different architectures?



**Summary Of The Paper:**

In this paper, the authors transplant the idea of spiking neurons to MLP.

**Summary Of The Review:**

In summary, this paper's attempt is worthy to be credited but it really lacks good motivation.

---

### Official Review · Reviewer_2Yab · 2022-10-28

**Confidence:** 4
**Correctness:** 3
**Technical Novelty And Significance:** 3
**Empirical Novelty And Significance:** 3
**Recommendation:** 3

**Clarity, Quality, Novelty And Reproducibility:**

The key problems this paper attempt to solve is not clear. There are already some works that exploit the combination of SNN and MLP, also, this paper doesn't provide new insights for this research direction. The proposed framework seems ordinary and regular. Some writing issues still exist in this paper, such as 'i.e' should be 'i.e.'
The authors promise to release the source code, and I believe the experiments can be re-produced if the code is available.

**Strength And Weaknesses:**

Strength
1. the direct training of deep SNN for energy-efficient classification is an interesting direction.



Weaknesses
1. the integration of SNN and MLP is not new in year 2022. [a]. Li, Wenshuo, et al. "Brain-inspired multilayer perceptron with spiking neurons." Proceedings of the IEEE/CVF Conference on Computer Vision and Pattern Recognition. 2022.
2. the key problems this paper solved for deep SNN are not clear. In another word, the novelty of this paper is limited.

**Summary Of The Paper:**

This paper proposes a MLP based spiking neural network, termed spiking MLP-Mixer, which contains the Spiking Token Block, Spiking Channel Block, and Speaking MLP. The authors state that they achieve good performance on the Image-1k dataset, Cifar10, Cifar100. This work suggests the importance of integrating global and local learning for optimal architecture design of SNN.




**Summary Of The Review:**

This paper proposes a MLP based spiking neural network, termed spiking MLP-Mixer, which contains the Spiking Token Block, Spiking Channel Block, and Speaking MLP. The authors state that they achieve good performance on the Image-1k dataset, Cifar10, Cifar100. This work suggests the importance of integrating global and local learning for optimal architecture design of SNN. But I think the proposed framework is not new, similar ideas can be found in previous works.

---

### Official Review · Reviewer_86ui · 2022-11-01

**Confidence:** 4
**Correctness:** 2
**Technical Novelty And Significance:** 2
**Empirical Novelty And Significance:** 2
**Recommendation:** 3

**Clarity, Quality, Novelty And Reproducibility:**

Clarity: Good.

Quality: Some comparisons and experiments are missing.

Novelty: Limited. Some part is hard to judge.

Reproducibility: Good.


**Strength And Weaknesses:**

Strengths:

1. Experiment results demonstrate competitive performance and energy efficiency compared with convolution-based SNNs.

2. Ablation study on skip connections and visualization of weights are studied to better understand the model.

Weaknesses:

1. The novelty and contributions of this paper are limited or hard to judge. There are two main designs in this paper: MLP-based design and spiking patch encoding module. The first one is directly adopted from recent ANN works, i.e. the recent popular patch-based methods such as transformers and MLP-Mixers. Replacing LN with BN is nothing new and has been studied in transformer architectures [1]. The design in this paper simply applies these techniques in ANNs to SNNs, without much specificity of SNNs. It is not clear what additional contributions this paper can bring. The second one, as said in the paper, is inspired from a recent work which is anonymous in the reference. However, this paper does not provide a copy manuscript in the supplementary material or discuss the difference in the main text, so it is hard to judge whether there is any novelty compared to that work.

2. Experiments do not compare the spiking MLP-Mixer with the ANN counterpart, regarding performance and energy efficiency.

3. There lacks comparison with some better results of previous SNN works, such as results with SEW-ResNet-34 architecture in [2,3] (67.04% and 68% on ImageNet), the state-of-the-art results on ImageNet with ResNet-34 architecture (68.19%) and VGG-16 architecture (71.24%) in [4], or the state-of-the-art results on CIFAR datasets in [5]. Since this paper focuses on network architecture design, the comparison with previous better architectures is required. The descriptions about the state-of-the-art results are not update-to-date.

Besides, since transformer-like architectures usually requires more training techniques such as strong data augmentations, this should also be discussed if this paper uses a stronger training setting than previous works.

4. There lacks ablation study on the proposed spiking patch encoding module. And also, since this module is inspired by the anonymous work, there should be comparison results to that work.

5. There is no experiment on dynamic datasets commonly used for SNNs such as DVS-Gesture or DVS-CIFAR10.

6. The authors should distinguish parenthetical and narrative types for references in the main text.

[1] Yao, Z., Cao, Y., Lin, Y., Liu, Z., Zhang, Z., & Hu, H. (2021). Leveraging batch normalization for vision transformers. In Proceedings of the IEEE/CVF International Conference on Computer Vision (pp. 413-422).

[2] Fang, W., Yu, Z., Chen, Y., Huang, T., Masquelier, T., & Tian, Y. (2021). Deep residual learning in spiking neural networks. Advances in Neural Information Processing Systems, 34, 21056-21069.

[3] Deng, S., Li, Y., Zhang, S., & Gu, S. (2021, September). Temporal Efficient Training of Spiking Neural Network via Gradient Re-weighting. In International Conference on Learning Representations.

[4] Li, Y., Guo, Y., Zhang, S., Deng, S., Hai, Y., & Gu, S. (2021). Differentiable spike: Rethinking gradient-descent for training spiking neural networks. Advances in Neural Information Processing Systems, 34, 23426-23439.

[5] Meng, Q., Xiao, M., Yan, S., Wang, Y., Lin, Z., & Luo, Z. Q. (2022). Training High-Performance Low-Latency Spiking Neural Networks by Differentiation on Spike Representation. In Proceedings of the IEEE/CVF Conference on Computer Vision and Pattern Recognition (pp. 12444-12453).

**Summary Of The Paper:**

This paper introduces recent MLP-based artificial neural network architecture design to spiking neural networks. By replacing the commonly used layer normalization in MLP-Mixer with batch normalization, this paper proposes spiking MLP-Mixer with multiplication-free inference. A spiking patch encoding module is also proposed to enhance local feature extraction in the model. Experiments on static image classification tasks demonstrate competitive performance and energy efficiency.

**Summary Of The Review:**

In summary, the experiment results are competitive, but the novelty and contributions are limited and there lack some important comparisons and experiments.

---

### Official Review · Reviewer_BQAF · 2022-11-01

**Confidence:** 5
**Correctness:** 3
**Technical Novelty And Significance:** 2
**Empirical Novelty And Significance:** 2
**Recommendation:** 3

**Clarity, Quality, Novelty And Reproducibility:**

+Paper is well written and easy to follow.
+ The contributions are clear and the results are good. I am not sure about novelty since it is a mix of methods that have existed in SNN/ANN literature and putting together seem to make the model better.
-I am concerned about whether this is truly advanategous SNN framework as I have raised questions around SNN implementation and the sparsity in weakness.

**Strength And Weaknesses:**

+ Very comprehensive results

+ Simple yet effective idea

- Since the authors use a BN technique, I am wondering if the authors can shed light on how their method differs from previous temporal BN methods proposed by prior works that have shown accuracy improvement while decreasing the total timesteps [1, 2].
[1]Kim, Y., & Panda, P. (2020). Revisiting batch normalization for training low-latency deep spiking neural networks from scratch. Frontiers in neuroscience, 1638.
[2] Zheng, Hanle, et al. "Going deeper with directly-trained larger spiking neural networks." Proceedings of the AAAI Conference on Artificial Intelligence. Vol. 35. No. 12. 2021.

- There is a plethora of works today on SNN algorithmic training-  precisely talking about how we can get improved accuracy with less timesteps. But, I am more concerned by the fact that in such large-scale settings, are SNNs going to be actually advantageous? The authors show some simplistic energy estimation results which is grossly approximate. For true energy estimation, they have to consider memory access and data access energy which turns out to expend a lot of computations in SNNs given their repeated time-wise computation. In a recent work [3], true energy estimation on a systolic accelerator precisely shows that SNNs are not very advantageous over ANNs because repeated timestep computations will lead to redundant access of weights and membrane potentials is going to further add to energy unless we really improve the sparsity rate. Can the authors kindly comment on this -  and it may be worthwhile for authors to include a discussion n the relevance of using more mainstream tools for energy estimation rather than just doing analytical modeling of FLOPS count?
[3] Yin, Ruokai, et al. "SATA: Sparsity-Aware Training Accelerator for Spiking Neural Networks." arXiv preprint arXiv:2204.05422 (2022).

- Coming to my next point, there is a recent work [4] that explores sparse SNNs using lottery ticket hypothesis to truly take advantage of SNNs energy efficiency over ANNs. Can the authors comment on how their model in terms of parameter count compares to these sparse SNN models which in fact show SOTA accuracy on CIFAR10,100 with very low timestep count?
[4] Kim, Youngeun, et al. "Lottery Ticket Hypothesis for Spiking Neural Networks." arXiv preprint arXiv:2207.01382 (2022).

- Finally, I think it is well known that SNNs will be more suited to DVS or event based tasks as compared to standard digital camera recognition models. Recent works have shown superiority of SNNs over ANNs on these neuromorphic datasets [5, 6, 7]. Can the authors run their model on one of these datasets and compare to [5,6,7]?
[5] Li, Yuhang, et al. "Neuromorphic Data Augmentation for Training Spiking Neural Networks." arXiv preprint arXiv:2203.06145 (2022).
[6] Kim, Youngeun, and Priyadarshini Panda. "Optimizing deeper spiking neural networks for dynamic vision sensing." Neural Networks 144 (2021): 686-698.
[7] Kim, Y., Chough, J., & Panda, P. (2022). Beyond classification: directly training spiking neural networks for semantic segmentation. Neuromorphic Computing and Engineering.

**Summary Of The Paper:**

The paper presents an interesting training algorithm for training SNNs from scratch. It uses a combination of skip connections and BN tools to get better accuracy on image recognition tasks.

**Summary Of The Review:**

Interesting paper, but as expressed in my concerns- i am not very convinced about the novelty and especially the energy efficiency advantages of the work.

---

### Comment · Area_Chair_3TAF · 2022-11-19
**Please respond to author rebuttal**

Dear Reviewers,

The authors have submitted their rebuttals. Please have a look and respond to their efforts. This will be a respect to their hard work. Many thanks!

Area Chair

---

### Decision · Program_Chairs · 2023-01-20

**Decision:**

Reject

**Justification For Why Not Higher Score:**

The paper should be rejected for sure.

**Justification For Why Not Lower Score:**

N/A

**Metareview: Summary, Strengths And Weaknesses:**

The paper originally got four 3s (reject) and one 5 (marginally below the threshold). The major challenges include limited novelty, unclear motivation, missing important related work and distuiguish from them, incomplete and unconvincing experimental results, etc. After rebuttal, one reviewer raised his/her score from 5 to 6. However, such scores do not warrant publication on ICLR. So the AC recommended rejection.


**Summary Of Ac-Reviewer Meeting:**

N/A